# Probing the symmetry of the potential of localized surface plasmon resonances with phase-shaped electron beams

Giulio Guzzinati[1], Armand Béché[1], Hugo Lourenço-Martins[2], Jérôme Martin[3], Mathieu Kociak[2] & Jo Verbeeck[1]

Plasmonics, the science and technology of the interaction of light with metallic objects, is fundamentally changing the way we can detect, generate and manipulate light. Although the field is progressing swiftly, thanks to the availability of nanoscale manufacturing and analysis methods, fundamental properties such as the plasmonic excitations' symmetries cannot be accessed directly, leading to a partial, sometimes incorrect, understanding of their properties. Here we overcome this limitation by deliberately shaping the wave function of an electron beam to match a plasmonic excitations' symmetry in a modified transmission electron microscope. We show experimentally and theoretically that this offers selective detection of specific plasmon modes within metallic nanoparticles, while excluding modes with other symmetries. This method resembles the widespread use of polarized light for the selective excitation of plasmon modes with the advantage of locally probing the response of individual plasmonic objects and a far wider range of symmetry selection criteria.

[1] EMAT, University of Antwerp, Groenenborgerlaan 171, 2020 Antwerp, Belgium. [2] Laboratoire de Physique des Solides, Univ. Paris-Sud, CNRS UMR 8502, F-91405 Orsay, France. [3] Institut Charles Delaunay—Laboratoire de nanotechnologies et d'instrumentation optique, UMR CNRS 6281, Université de Technologie de Troyes, 10010 Troyes, France. Correspondence and requests for materials should be addressed to G.G. (email: giulio.guzzinati@uantwerpen.be).

Plasmon resonances are collective excitations of the conduction electrons in metals. In particular in metallic nanoparticles, the confinement of the conduction electron gas causes the appearance of self-sustaining resonances known as localized surface plasmon resonances (SPRs). The extraordinary properties of SPR, such as strong local electrical fields, dramatic spatial variations over a single particle or high sensitivity to nanometre scale environmental changes, offer an attractive way to perform sub-wavelength manipulation of electromagnetic waves in the infrared to ultraviolet range.

The study of these phenomena has wide and far-reaching consequences. A new generation of opto-electronic devices is being developed by manipulating visible light waves (wavelengths between 390 and 700 nm) with the same methods used in radio technology (commonly used wavelengths range from several metres down to millimetres)[1] and metamaterials are showing just how far the tailoring of optical properties can go. New and remarkable examples of this are continuously emerging, ranging from on-chip light spectrometers and linear accelerators[2,3], plasmonic rectennas[4], increased efficiency LED (light emitting diode) and photovoltaics[5], negative refractive index and slow-light materials[6,7]. Other applications of plasmonics are also abundant, including surface-enhanced Raman spectroscopy[8] and cancer therapy[9].

This progress has been enabled, among other things, by the availability of nanoscale characterization techniques, such as optical spectroscopies, scanning near-field optical microscopy or electron spectroscopies in the transmission electron microscope (TEM). The latter are undergoing significant development recently, as the now conventional electron energy loss spectroscopy (EELS)[10] is being supplemented by electron energy gain spectroscopy[11,12] and cathodoluminescence[13], whereas photon-induced near-field electron microscopy is opening the possibility to image the time evolution of the resonances down to the sub-picosecond range[14,15].

Despite the power of these techniques, many aspects in the behaviour of plasmon resonances cannot be probed directly. Although optical spectroscopies can measure the response of the nanostructures to the different polarizations of light, they lack the spatial resolution to directly study the local field distribution around a single nanoparticle and even scanning near-field optical microscopy only allows to study in detail the largest structures. Although beam shaping has successfully been used in light optics to offer an extra degree of control over the response of plasmonic systems (see ref. 16 for an interesting parallel to the present work), the spatial resolution remains limited. Electron spectroscopies allow to probe the particles with nanometre resolution and the point-charge character of the fast electrons in the beam excites (and probes) modes of all multipolarities, as well as dark modes. The existing electron spectroscopy techniques however all probe (schematically) the projected electromagnetic local density of states, which is proportional to the square of the local electric field $I \propto |E_z|^2$, and are therefore blind to the sign or the phase of the field. Applications relying on the detailed knowledge of the field phase can only retrieve it through simulations and devices can only be diagnosed through overall performance rather than allowing a fine-grained analysis. Furthermore, it is impossible to distinguish modes whose electrical field modulus ($|E_z|$) and energy are near degenerate, a task which can get especially difficult in high-symmetry systems. In the lower symmetry cases, even the simulations become of limited help as the computation time increases greatly.

Here we show how the local potential in plasmonic resonances couples to the phase of an electron beam through inelastic scattering. Furthermore, we suggest that the deliberate modulation of the phase of the incident electron beam can provide access to the sign of the plasmonic potential, entirely lifting these limitations. Finally, we demonstrate this by showing the selective detection of dipolar excitations by using a two-lobed beam reproducing its symmetry.

## Results

**A schematic representation of the experimental geometry**. The system we study, is schematically represented in Fig. 1. The inelastic interaction between an electron beam and a plasmonic resonance imprints the former with a phase proportional to the latter's local potential (Fig. 1a–c). When this phase is imposed on a shaped electron beam, specific selection rules appear in the transmitted direction and only resonances that possess the same symmetry as the impinging beam are expected to be detected (Fig. 1d) along the optical axis. In the following, we first give the theoretical justification of this heuristic guess and reformulate this problem in a concise expression showing that the problem exposed here shares a one-to-one correspondence with the problem of optical transitions in atomic physics. We then experimentally demonstrate the technique by using a two-lobed Hermite-Gaussian-like beam to selectively and directionally detect dipolar resonances in a plasmonic nanorod, analogously to what is currently possible with linearly polarized light, but now also providing local information in the deep sub-wavelength range. Finally, we show how the process can be generalized to arbitrary symmetry, with the example of quasi-degenerate quadrupolar and dipolar modes in squared plasmonic particle.

**Theoretical treatment**. To understand the role of the phase in the inelastic interactions between an electron wave and an SPR, a new semiclassical description of the process is necessary. In the following, the SPR modes will be described as a set of standing waves, indexed by a number $m$. For simplicity, the description will be given within the quasistatic approximation, which permits a straightforward interpretation while preserving the salient physical features of our findings. In this approximation, each mode $m$ is associated to an eigen(electric)potential $\phi_m(\mathbf{r})$, which depends only on the geometry of the particle subtending the plasmon, and a spectral function $g_m(\omega)$[17,18], which depends both on the geometry and on the dielectric constant of the constituting material. In the case of a metal, the imaginary part of each $g_m(\omega)$ is typically described by a lorentzian peaking at the SPR resonant energy[17,18]. All the spatial and spectral features of SPR are therefore totally unveiled once $g_m(\omega)$ and $\phi_m(\mathbf{r})$ are known. These quantities can be straightforwardly simulated[17,19,20].

An incoming electron beam in the initial state $\Psi_i(\mathbf{r})$ will exchange energy and momentum with the plasmon, therefore being transformed into a final state $\Psi_f(\mathbf{r})$. A common assumption in electron microscopy[21,22] (see Methods), which is well justified in the present case, is to write the electron wavefunction as the product of a component describing the motion parallel to the optical $z$ axis with a component $\Psi_\perp(x, y)$ containing the modulations in the transverse plane $(x, y)$:

$$\Psi(\mathbf{r}) \propto \Psi_\perp(x, y) \exp ik_z z, \qquad (1)$$

where $k_z$ is the electron wave-vector component along the optical axis.

Under these conditions, the probability density for energy loss $\hbar\omega$ of an electron travelling with velocity $v$ can be expressed in the form of a transition matrix element (see Supplementary Note 1):

$$\Gamma(\omega) = \frac{2e^2}{hv^2} \sum_m \sum_{f,\perp} \Im\{-g_m(\omega)\} \cdot \left| \int \Psi_{f,\perp}(x, y) \tilde{\phi}_m(x, y, q_z) \Psi_{i,\perp}^*(x, y) \, dxdy \right|^2,$$

$$(2)$$

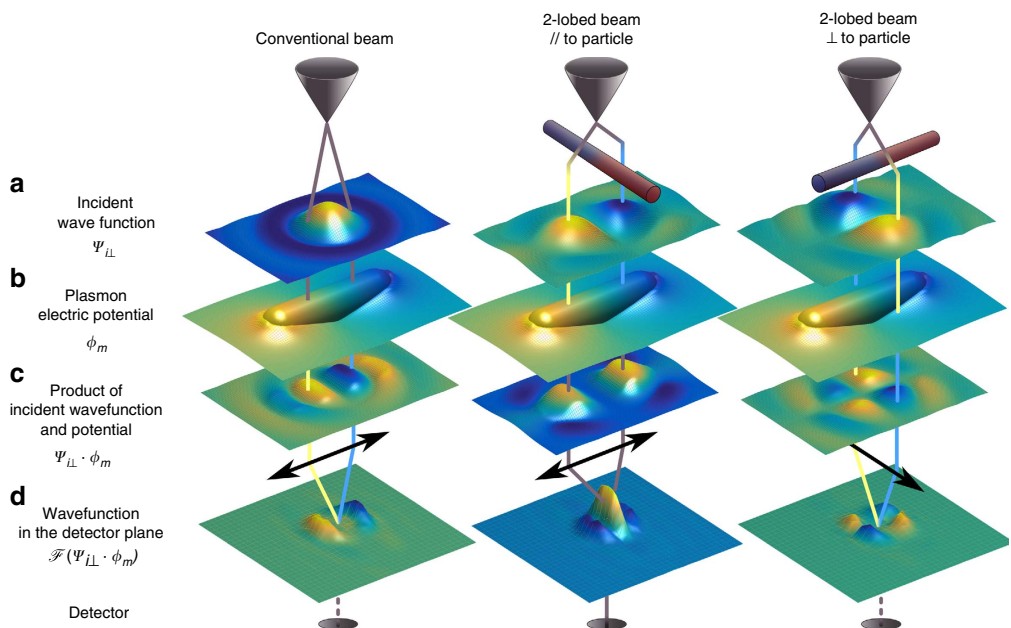

**Figure 1 | Schematic and simplified representation of the experiment.** The columns describe the propagation of three different electron wave functions. The first one is a conventional beam with constant phase surfaces. The second and third correspond to two-lobed beams as generated by a particular phase plate (see Fig. 2). The wave function of an electron (**a**) inelastically interacts with a (dipolar) plasmonic resonance in a metallic nanorod (**b**). The coherent interaction between the beam's in-plane wave function and the plasmon's potential determines the relative phase of the different rays (see blue/yellow or grey ray pairs) and whether they interfere constructively or destructively in the forward direction in the detection plane (**d**). A spectrometer only accepting electrons scattered in the centre of the detection plane (dark ellipses at the bottom) is then sensitive only to plasmon modes, which match the symmetry of the probe (second column), and ignores the other cases (first and third column) where electrons are only scattered off axis. The interaction is here simplified with $\Psi_{i\perp} \cdot \phi_m$ in **c**, but is actually given by equation (2)). For **a,c,d**, the height in the three-dimensional representation is proportional to $|\Psi_\perp(r_\perp)|^2$ and the colour $\propto \Psi(r_\perp)$, whereas for **b** colour indicates the plasmon's electric potential in the mid plane of the particle. In **d**, $\mathscr{F}$ represents the Fourier transform, as the measurement is performed in the far field.

where $\tilde{\phi}_m(x,y,q_z)$ is the Fourier transform along $z$ of the eigenpotential, and $e$ and $h$ are respectively the elementary charge and the Planck constant. The expression under the modulus square closely resembles an atomic transition probability, where the free electron states correspond to the bound atomic states, transiting under an external perturbation here replaced by the plasmonic potential. This is in line with the usual analogy made between phase-shaped free electron beams and atomic states in free space[23] and extends it with the possibility of transitions.

More interestingly, this view of equation (2) can be reversed pointing to its main application: unlike atomic states, it is possible to shape and manipulate free electron states, for instance by giving them a definite symmetry, and thus allowing for complete analysis of the symmetries of the plasmonic potential. This is the essence of the present work.

Indeed electron beams with a tailored phase profile and their unique properties have gathered a significant amount of attention in recent years[24,25]. After the first demonstration of electron vortex beams[26–28], great steps have been made in the methods for controlling the electron's phase[29–34], leading to the demonstrations of new electron beam types such as Airy waves or Bessel beams[34–36], and several suggestions for possible applications[37–41]. In particular, proposals have been made to use vortex beams for dichroic measurement on chiral resonances of plasmonic structures[22] or to match the rotational symmetry of higher order beams with the one of higher multipolarity plasmonic resonances[42]. Experimental results however have yet to follow due to the high complexity of the experiments involved. The generality of the formalism outlined in this work allows to develop new experiments, with a simpler design and a straightforward physical interpretation.

To explore the applications of modified beams, the next step is then to insert test wave functions in equation (2) and observe how the symmetry of the beam couples to the one of the eigenpotential. Starting from the simplest case, the dipolar excitation of a metallic nanorod (as in Fig. 1b), an interesting choice for the incoming wave-function is one which possesses the same symmetry as the plasmon's eigenpotential, that is, formed by two lobes opposite in phase (see Fig. 1a). Following experimental constraint, it is also worth considering only transitions to a final plane wave state. This simple case is particularly relevant, as most plasmonic excitations exhibit alternating lobes of charge density. Therefore, this type of 'dipolar' beam can locally match plasmons of arbitrary symmetries and can be used as an universal symmetry probe.

Simple examples of such states are:

$$\Psi_{i,\perp}(x,y) \propto x \exp\left(-\frac{x^2+y^2}{w^2}\right) \tag{3}$$

$$\Psi_{f,\perp}(x,y) = \text{constant}, \tag{4}$$

where $w$ is a sizing parameter. Equation (2) then becomes:

$$\Gamma(\omega) \propto \sum_m \mathfrak{J}\{-g_m(\omega)\} \cdot \left| \int x \exp\left(-\frac{x^2+y^2}{w^2}\right) \tilde{\phi}_m(x,y,q_z)\,\mathrm{d}x\mathrm{d}y \right|^2. \tag{5}$$

As the particle under study is a rod, its plasmon resonances possess a definite symmetry, with their eigenpotentials being either symmetric or antisymmetric[18] (as in Fig. 3c), which has important repercussions here. The odd symmetry of $x$ makes the integral vanish for any excitation possessing an even

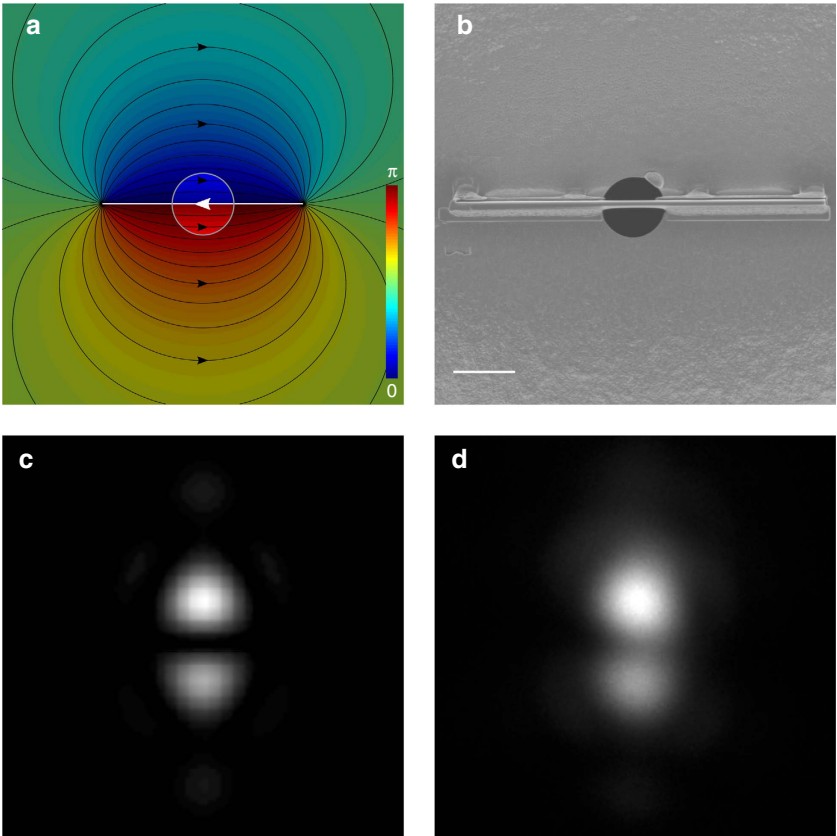

**Figure 2 | Generation of Hermite-Gaussian like beams with a magnetic phase plate. (a)** The magnetic field lines (black lines) and the corresponding Aharonov–Bohm phase shift (colour hue) induced by a uniformly magnetized straight needle. Isolating with a round aperture (grey circle) a small area near the centre of the needle, results in two semi-circles with a nearly uniform phase. **(b)** The practical realization of this setup, by placing a microscopic ferromagnetic needle across a round aperture, giving a phase shift of $0.87\pi$ (see Supplementary Note 3). Scale bar, 5 μm. **(c,d)** Simulated and experimental beam intensity profiles. The simulation assumes a phase difference equal to the experimental value of $0.87 \cdot \pi$ rad, and displays an equivalent asymmetry.

eigenpotential. Antisymmetrical ones on the other hand can contribute to the transition depending on the amount of the overlap between $\Psi_i$ and $\phi_m$, and the strongest effect is obtained for a dipolar excitation and for a value of $w$ that approaches the size of the nanorod, showing the intended selectivity (Fig. 1, second column). It is also worth noting that once these conditions are satisfied, the transition probability is maximum for a plane-wave final state propagating along the optical axis and decreases rapidly for tilted plane waves ($\Psi_{f,\perp}(x,y) \propto \exp\left[i\left(k_x x + k_y y\right)\right]$), where $k_x$ and $k_y$ are the transverse momentum components) as can be intuitively understood in Fig. 1d.

Conversely, for a conventional unstructured incident electron beam centred in the middle of a nanoantenna (for example, a Gaussian beam $\Psi_{i,\perp}(x,y) \propto \exp\left(-(x^2+y^2)/w^2\right)$), the transition probability to a final on-axis plane wave state cancels out for all odd eigenpotentials, but transitions to tilted waves are possible (as in the first column of Fig. 1).

This crucial observation reveals that an effective detection of these new effects requires a projective measurement, done by only analysing the electrons that travel very close to the optical axis and thus by limiting the acceptance angle of our detector.

Interestingly, this set-up can also be interpreted as a two arm interferometer (see Fig. 1). The relative phase difference acquired by two ray paths, either due to the beam modification or to the interaction with the plasmonic resonance, determines the constructive or destructive interference in the detection plane. Complete constructive interference is obtained if the phases given by beam shaping and plasmon cancel each other out.

It is important to point out that without a projective measurement, that is, a limited collection angle, the selectivity disappears. Transitions to tilted plane waves, as noted above, are still allowed and in general for electron beams a change in the phase alone will not modify the total cross-section but only the angular distribution of scattered electron, in contrast to what happens in light optics upon a change in polarization. A beam with identical intensity distribution to the two-lobed beam of equation (3), but without the sign change ($\Psi_{i,\perp}(x,y) \propto |x| \exp\left(-(x^2+y^2)/w^2\right)$) would result in an identical total cross-section if all possible final states are allowed. It is only by performing an angular post-selection (that is, measuring on axis) that the two beams give different selection rules.

**Manipulation of the beam's wave function.** With these new insights, an appropriate experimental setup to test these predictions can be prepared. The first element needed is an efficient way to generate a two-lobed beam such as the one presented in equation (3). The simplest approach is to neglect the intensity modulation, imprint a beam with the characteristic phase required through a specially modified beam-limiting aperture and then employ the far field diffraction pattern of this aperture as a probe. In this case, the two halves of the aperture should be opposite in phase. Although ways to do this have been suggested and demonstrated before[31,43,44], none of these is practical in our case. Indeed, computer-generated holograms produce multiple beams that simultaneously interact

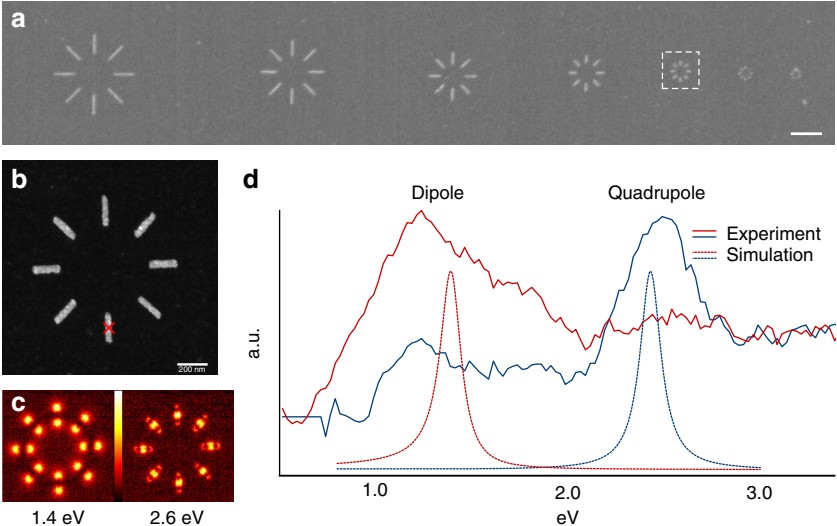

**Figure 3 | Sample overview and experimental spectra.** We prepared a sample (**a**) comprising aluminium nanorods of different lengths and orientations (scale bar, 1 µm (**a**)). The structures display very clear plasmonic modes, as in the case of the dipolar and quadrupolar resonances of the 200 nm rods displayed here (**b,c**). We collected EELS spectra by placing electron beams on the centre of one of these rods (marked in red in **b**). The spectra acquired (**d**) using the beam generated by the modified aperture (oriented parallel to the rod as in the second column of Fig. 1) and a conventional beam produced with a round aperture of the same size, display strong modal selectivity (solid lines), in agreement with our expectations (see first two columns of Fig. 1) and with our numerical simulations (dashed lines). The spectra have been scaled for ease of visualization as the blue spectra were significantly less intense, in agreement with the expected lower intensity of a quadrupolar peak.

with the sample, whereas phase plates based on the thickness of the material suffer from charging, contamination and their use is limited to one specific value of kinetic energy of the electron beam.

To avoid these problems we took a different approach based on the Aharonov–Bohm effect[30,45]. When two electronic paths enclose a magnetic flux $\Phi_B$, the two paths acquire a phase difference proportional to the flux $\Delta\phi = -e\Phi_B/\hbar$[46,47]. In the ideal geometry this flux would be localized in one infinitesimally thin and infinitely long flux line, crossing the aperture across its diameter and dividing it in two equal parts characterized by a uniform phase differing by $\pi$ radians[34].

A good approximation to this can be realized with a microscopic ferromagnetic needle of length much larger than the aperture, with carefully controlled and uniform cross section. In such an object, the magnetization is constrained in the needle's long direction due to shape anisotropy and the uniform cross-section ensures that no field lines emerge from the sides (Fig. 2a). When this needle is placed with its centre across a circular aperture (Fig. 2b), its ends appear far and the flux given by the returning field lines can be neglected with respect to the high flux density present inside the needle. Under these conditions, the magnetized bar separates the aperture in two parts, causing a constant phase difference between the two halves (Fig. 2a).

The actual fabrication of such a needle is challenging and the control over the phase difference is limited. Nevertheless, it was possible to reach a phase shift of $0.87\pi$ rad, very close to the desired one of $\pi$ radians.

This aperture was then inserted in the illumination system of a TEM, generating the probe intensity profile shown in Fig. 2d. As the plane where the aperture is placed is optically reciprocal to the sample plane, imperfections in the desired phase profile also cause the intensity distribution to change. Owing to the phase difference not being exactly $\pi$ radians, the beam is slightly asymmetric and presents a more intense and a weaker lobe, as is also well reproduced in numerical simulations performed using the experimental value of the phase difference (Fig. 2c) (see also

the Supplementary Note 4). As desired, the probe is formed of two intensity lobes separated by a dark line.

**Spectroscopic experiments**. As a test sample with strong plasmonic response in the visible or near infrared optical range, we choose to study aluminium nanorods[48] realized through electron lithography (see Methods).

When analysed with conventional EELS, the nanorods exhibited clear and marked plasmonic resonances as shown in Fig. 3c. Although conventional scanning TEM–EELS mapping can not measure the charge sign, in this simple system it is easy to deduct the different charge symmetries of the two resonances displayed here. The lower energy mode corresponds to a dipolar charge oscillation (two maxima separated by a minimum), whereas the other corresponds to a quadrupolar charge configuration (the central maximum being twice as intense as the ones on the tips).

To obtain the highest overlap between the beam and the plasmonic excitation the probe size was maximized, reaching about 50 nm in the long direction. In addition, the semi-collection angle was reduced to limit the detection to plane waves propagating along the optical axis. Since, as can be seen from Fig. 3c, the centre of the rod position coincides with an intensity minimum of the dipolar mode, but with a maximum for the quadrupolar one, a conventional spectroscopic experiment would detect a very weak signal from the former and a much stronger one from the latter. If the phase modified beam is employed, on the other hand, the opposite situation is expected, with a very strong signal from the dipolar excitation and little to none from the higher order one.

The loss spectra collected using a conventional round aperture and the modified aperture are displayed in Fig. 3d. As can be observed, the two spectra display strong selectivity in agreement with the above arguments, with the imperfect selection being due to the finiteness of the collection angle.

**Figure 4 | Orientation and position dependence of the signal given by the modified beam.** We studied the position dependence of the signal collected with our modified beam (represented in **a** in yellow and blue colours) and corresponding to the dipolar resonance of a 400 nm rod (scale bar 100 nm (**a**)). This corresponds to studying the intensity of the dipolar peak (red peak in Fig. 3d) for a variety of different beam positions, as schematically shown in **a**. For all the imperfections of these early experiments, there is remarkable agreement between the simulated (**b,d**) and experimental data (**c,e**). If the probe is parallel to the rod, the map shows a strong peak in the centre, with two satellite lobes on the side, for a probe oriented orthogonally to the particle, the centre shows a strong dip with four satellite lobes. It is interesting to observe how the signal intensity seems to correlate to intensity variation of the plasmon field along the probe orientation.

To further explore and understand these results, a numerical simulation code was developed, based on integrating equation (5) with numerically simulated eigenmodes and beam profiles (see Methods). The resulting spectra (dashed lines in Fig. 3d) show the same type of selectivity as the experimental ones. The slight mismatch in the excitation energies is entirely expected, due to the simplified model used to represent the sample.

Further insight can be obtained by studying the effects of alignment and azimuthal angle between beam and particle. We simulated the transition probability induced by the dipolar mode of a 400 nm nanorod on our modified electron beam, for two different orientations of the probe, either parallel or perpendicular to the rod (reproduced in Fig. 4b,d). The simulations show that the transition probability for a beam centred on the particle will be maximized when parallel and vanish when orthogonal, allowing to directionally probe the plasmonic response in a manner similar to that of linearly polarized light and in agreement with the symmetry arguments shown in Fig. 1. When shifting the beam position (see Methods), the collected signal appears sensitive to directional variations in the plasmon's intensity. An experimental acquisition of the same maps, despite the artefacts induced by the beam's asymmetry and the slight error over the azimuthal angle, appear in good qualitative agreement. The map acquired with the probe oriented parallel to the rod, shows a central maximum and two side lobes (Fig. 4c), whereas the other one, acquired with the probe orthogonal to the nanorod, possesses a structure presenting a central minimum surrounded by four peripheral intensity lobes (Fig. 4e).

## Discussion

Here we have shown how the phase of an electron beam couples to the real potential of localized SPRs and how this coupling can be exploited with a deliberate manipulation of the electron beam's wave function plus a post selection of the electrons scattered on axis. We have experimentally verified this by using a two-lobed beam to selectively probe the dipolar resonances of a nanorod. Although we have shown how this Hermite–Gaussian-like beam can be used as an electron beam analogue of linearly polarized light and is also able to locally and directionally probe the plasmonic field, this is only a first example of the many potential application of this approach. For instance, crossing orthogonally two needles such as the one used here over a round aperture would generate a four-lobed beam,

analogous to a $HG_{11}$ Hermite–Gaussian mode, which would couple to the quadrupolar modes of a cubic or square particle. Unlike for the nanorod, in such high-symmetry particles the conventional techniques do not allow to clearly distinguish the charge multipolarity of the different resonances and the EELS maps of both the dipolar and quadrupolar mode appear fourfold symmetric. Phase-shaped beams, on the contrary, allow to address the charge symmetry in a direct way. As an example we simulated (Fig. 5) the response of a 100 nm × 100 nm × 20 nm silver square prism. Although conventional EELS maps of the first two plasmonic excitations (Fig. 5b) appear very similar (in real experiments, the difference is hardly detectable[49]) and a conventional beam will couple to both modes, a two-lobed and a four-lobed modified beams will couple only to the mode possessing the same symmetry, thus giving away the first mode as dipolar and the second one as quadrupolar (Fig. 5c). Furthermore, the two-lobed beams can be used, depending on the orientation, to separately probe the two degenerate dipolar mode (in the inset of Fig. 5c).

Another interesting example is that of coupled nanoparticles. The hybridization of the plasmon modes in sets of nanostructures at a short distance causes the original modes to split into several modes of different symmetry with shifted energies[50]. Depending on the distance between the particles (and thus the coupling) and the number of nanoparticles, the energy shift can be below the energy resolution of conventional EELS, whereas shaped beams can exploit the difference in symmetry to excite different modes separately (see Supplementary Note 5).

This shows how the approach in itself is entirely general. Although currently electron phase manipulation is cumbersome and difficult, our results show that arbitrary wave shaping would open up a tremendous potential for tuned plasmonic measurements in the TEM, allowing to adapt the probe shape to the property to be measured, augmenting its already wide possibilities with information previously either not available on a local basis or completely out of reach.

## Methods

**Aperture production.** The magnetized needle has been prepared out of a 60 nm thick nickel film by focused ion beam (FIB) and then placed across a 5 μm aperture, also milled by FIB out of a beam-opaque platinum film. After the preparation, the phase shift is measured through electron holography (see Supplementary Note 3) and the needle thickness is further refined through FIB, to best approximate the desired phase shift. The accuracy in determining the phase shift is limited by the

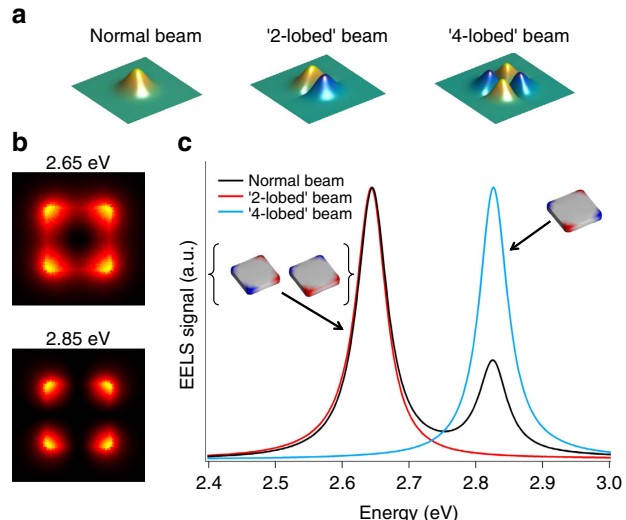

**Figure 5 | Use of modified beams to detect charge multipolarity.** Different beam shapes (**a**) couple to different modes. In the case of a high symmetry particle such as a metallic square prism, conventional EELS maps (simulated in **b**) present the same symmetry, and the multipolarity can't be evaluated. Spectra have been simulated (**c**) for different beams centred on the square particle, and while the conventional probe detects all mode, it can be observed how the modified beams couple selectively to one of the two modes, revealing its charge symmetry.

machining resolution achievable by FIB and by the reduction in the magnetic properties of the needle due to the ion bombardment.

**Sample production.** Aluminium nanorods have been fabricated using electron beam lithography in a FEG SEM system (eLine, Raith). First, a 150 nm-thick layer of resist of resist of poly(methyl methacrylate) was spin-coated on a scanning TEM–EELS compatible substrate. The latter consists in arrays of 15 nm-thick $Si_3N_4$ square membranes engraved in a small silicon wafer (3 mm diameter). The membranes were subsequently impressed by the electron beam using the EBL system (doses varying between 150 and 300 $\mu C\,cm^{-2}$). The patterns in the resist were then developed for 60 s in a 1:3 MIBK:IPA solution at room temperature. Then, a 40 nm-thick layer of Al was deposited on the sample using thermal evaporation (ME300, Plassys). Finally, lift-off has been accomplished by immersing the sample in acetone unveiling the Al structures on the membranes. The width of the nanorods, approximately uniform, is between 40 and 50 nm.

**Spectroscopic experiments and data treatment.** The spectroscopic experiments have been performed on a FEI Titan[3] TEM, outfitted with a Wien filter monochromator, an aberration-corrected illumination system and a Gatan Enfinium electron spectrometer. To reach the low angles required by these experiments, we operated the microscope in low magnification mode, switching almost completely off the objective lens. The spectra in Fig. 3 have been acquired employing an acceleration voltage of 120 kV and a semi-collection angle of ∼20 μrad, whereas the maps in Fig. 3 have been acquired with a high tension of 60 kV and a semi-collection angle of about 10 μrad. In both cases the illumination semi-convergence angle is ∼20 μrad.

The spectra presented in Fig. 3 have been treated by normalizing them and then subtracting reference zero loss peaks. The zero-loss peaks have been recorded immediately before the spectrum acquisition, with the corresponding aperture and by placing the beam over a hole in the sample to eliminate any spectral feature corresponding to the sample.

**Numerical simulations.** The simulation code was developed on top of the freely available package MNPBEM, a boundary element method simulation code. First, eigenmodes were numerically computed using MNPBEM[19,20]. Once we obtained the eigenmodes and spectral functions, we integrated equation (2) using for $\Psi_{i,\perp}$ a numerically simulated probe profile.

**Data availability.** All relevant data are available from the authors.

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

## Acknowledgements

We thank F.J. García de Abajo and D.M. Ugarte for interesting and fruitful discussion. This work was supported by funding from the European Research Council under the 7th Framework Program (FP7) ERC Starting Grant 278510 VORTEX. Financial support from the European Union under the Framework 7 program under a contract for an Integrated Infrastructure Initiative (Reference number 312483 ESTEEM2) is also gratefully acknowledged. Aluminum nanostructures were fabricated using the Nanomat nanofabrication facility.

## Author contributions

G.G., M.K. and J.V. conceived the experiment and designed the sample, which was fabricated by J.M. M.K. developed the theory and H.L.-M. wrote the related simulation code and performed the numerical simulations. M.K. and H.L.-M. verified the consistency of analytical calculations and simulations. A.B. manufactured the TEM aperture. G.G., A.B., M.K. and J.V. designed the experimental set-up, and G.G. performed the spectroscopic experiments and analysed the data. All authors contributed to writing the paper.

## Additional information

**Competing interests:** The authors declare no competing financial interests.

**Publisher's note**: 

