## [Peer Review File · Nature Communications]

Reviewers' comments:

Reviewer #1 (Remarks to the Author):

Report on: "Probing the symmetry and phase of localised surface plasmon resonances with phase-shaped electron beams » by Giulio Guzzinati, Armand B  ch  , Hugo Louren  o- Martins, J  rome Martin, Mathieu Kociak, and Jo Verbeeck

Ref: # NCOMMS-16-24651-T

This paper reports on the use of phase-shaped electron beam to selectively probe by TEM the symmetries of plasmonic resonances in metallic nanostructures. The main idea of the paper consists creating a 2-lobes electron beam whose phase symmetry allows discriminating the symmetries of the surface plasmon resonances (SPR) of a given metallic nano-object. To achieve the phase shaping of the incident electron beam, the authors use a ferromagnetic nanowire located in the condenser aperture. Such very simple device, similar to one already the used by the group in Antwerp for EMCD studies, allows creating two lobes beams with almost π phase shift difference between them. The interaction between this two phase-shifted lobes beam with the SPR is either constructive or destructive depending on the relative phases of the beam and the SPR symmetry. This simple and smart idea is analogous to polarised light / SPR interactions but offers the advantage to locally probe the SPR symmetry of individual object, thanks to the spatial resolution of TEM,

The paper clearly describes the method both theoretical and experimentally. Some points however are not so clear and could be improved for the benefit of the reader.

- The paper is first introduced by a schematic view of the experiment (fig. 1), which illustrates the effects of a conventional and the 2-lobed beam when interacting with the dipolar plasmonic resonance of a metallic nanorod. The illustration regarding the case where the 2-lobed beam is set perpendicular to the main axis of the Al nanorod is not so clear for me as I would expect almost no interaction with the dipolar SPR. Does the result presented in Fig. 1 (d) (right column) correspond to the interaction of the 2-lobed beam with the quadrupolar SPR of the nanorod (which is not represented neither in the figure nor in the text)? If yes I suggest the authors to clarify this point, if not I also suggest the authors to explain what is the expected result of the 2-lobed beam with the dipolar SPR in such configuration.
- The 2-lobed beam is created thanks to the use of magnetic nanowires magnetized along the wire direction resulting in a phase shift between the beams passing on each side of the wire thanks to the Aharonov –Bohm effect. In addition to the phase shift (which depends on the magnetic induction within the wire), the authors evidence an intensity difference between these two beams and claim it comes from a non perfectly equal to π phase shift (0.87π). This is not straightforward for me, as we do not expect to observe any difference in the intensity of the two parts of the beam whatever the phase shift between them. Could the intensity difference originate from a slight shift of the incident beam with respect to the magnetic nanowire above the condenser aperture ? Can the authors also clarify this point.
- In figure 3, the authors present experimental and simulated EELS spectra obtained on dipolar and quadrupolar SPR of nanorods whose orientation is varied compared to the axis of the 2-lobed shaped beam (blue curves) and for normal round beam (red curves). The experimental spectra are very convincing on the influence of the beam symmetry with respect to the symmetry of the SPR. However, the simulated plot with a "normal" beam only reveals the quadrupole mode at 2.6 eV (dashed blue plot) while the dipolar one at 1.4 eV is also experimentally observed and expected?
- At the end of the paper the authors show by simulation how a 2-lobed beam and even a 4-lobed beam could be efficient for probing the degenerated symmetries of SPRs in high symmetry particles like metallic square platelet (figure 5) or to analyse the coupling of plasmon modes in neighbouring nanoparticles. As a practical point of view and by curiosity, I wonder how the authors plan to set a 4-lobed beam as crossing 2 magnetic nanowires in the aperture condenser will result by dipolar magnetic interaction in a magnetic configuration quite far from the simple addition of

two independent magnetic nanowire magnetized along their axis?

To summarize, the main purpose of the paper demonstrating a smart way to probe the symmetries of SPR as well as its experimental and theoretical support largely deserve to be published in Nature Communications after the few previous points have been discussed and corrected.

Reviewer #2 (Remarks to the Author):

Review on "probing the symmetry and phase of localised surface plasmon resonances with phase-shaped electron beams":

In their manuscript entitled "Probing the symmetry and phase of localised surface plasmon resonances with phase-shaped electron beams", Guzzinati et al demonstrate theoretically and experimentally that using phase-shaped electron beams in a TEM allows detecting specific surface plasmon modes based on their symmetry. Although the concept bears great similarity with the use of polarized light in optical microscopes to selectively excite specific plasmon modes, the presented concept goes beyond as it adds the nanoscale spatial resolution of TEM.

This work brings important and new results. The paper is extremely well written. In particular, the introductory discussion and presentation of this new concept is very clear. Both theoretical and experimental results are very convincing. The results presented in Figure 4 clearly lie at the limit of what can be achieved today on state of the art Transmission Electron Microscopes.

In conclusion, this paper proposes and demonstrates a new concept that will undoubtedly be of the highest interest for the nano-optics community. The presentation of the results and their discussion is very clear. I strongly recommend this paper for publication in Nature Communications.

I have two comments:

1) Contrary to what is suggested in the title, the phase of the plasmon modes is only indirectly probed in the described experiments. In their work, the authors demonstrate that an a-priori knowledge of the symmetries of the plasmon modes supported by a metal nanoparticle allows choosing the electron beam shape that can selectively detect a given mode. However, the presented technique does not yield a quantitative measurement of the phase of the plasmon mode but rather gives access to its symmetry which is already an excellent step forward. Using the analogy with atomic physics, information about the symmetry of the coupling operator can be gained from the selection rules but no information about the phase.

2) In the present work, the authors show how the introduction of a phase jump in the wavefunction of an electron beam can excite selected plasmon modes. This strategy can be compared to the introduction of phase jumps in the incident electric field driving a gap nanoantenna as proposed in Volpe et al, Nano Lett., 2009, 9 (10), pp 3608–3611. In this work, high order Hermite-Gaussian beams have been used to switch on and off the hot spots of a linear dimer gold nanotenna. This work could be added as a reference.

3) The quality of some figures could be improved (fonts homogeneized for instance)

4) Line 281: Are the spectra normalized before or after substraction of reference zero loss peaks ?

Additional remarks:

- 1) Figure 1 - caption: the height in the 3D representation is proportional to the modulus squared of ψ , isn't it ?
- 2) Figure 1 - caption: it would be better to use the same font for F as in the figure.
- 3) Line 43 : "electric field"
- 4) Line 80: Is it "present" or "presented" ?
- 5) Figure 5 - Caption: "with our modified and corresponding". A word is missing
- 6) Line 210: "for two different orientationS". An "s" is missing.
- 7) Line 234: "Are appear very".
- 8) Line 285 and 287 have a repetition "using the freely available simulation code MNPBEM"
- 9) Line 288: "eigenmodes and spectral functions". An "s" is missing.
- 10) Line 428: "we need a to specify"
- 11) Line 431: "the opticAL axis"
- 12) Line 433: "we can sum, and due to the non-recoil approximation, resul...". This sentence should be improved.

Reviewer #3 (Remarks to the Author):

The authors study the plasmonic response of aluminum nanorods using electron energy-loss spectroscopy (EELS) in a modified transmission electron microscope. By generating an electron beam with a Hermite-Gaussian phase distribution, the authors are able to selectively excite plasmonic modes of certain symmetry. In addition, the authors can also retrieve phase information of the plasmonic mode, which is not possible with standard EELS (i.e., non-phased shaped beams).

I find the idea of using phase-shaped electron beams in EELS for optical studies very promising and appealing for a broad range of applications. I applaud the authors for the significant novel improvement on this topic shown in their manuscript, but I find the experimental results slightly demotivating. I think this is mainly rooted in the nanorod structure, which was chosen by the authors to highlight the benefit of phased-shaped beams. While Fig. 3d shows that a shaped beam and a normal beam give rise to resonances at different energies (due to excitation of dipole and quadrupolar modes, respectively) for one particular position, the data shown in Figure 4 is less convincing (especially Fig. 4c). Given the quite large size of the nanorod in Fig. 4, the data suggests that when employing a phase-shaped beam you lose the beneficial spatial resolution of standard EELS (which is also praised by the authors in their introduction). I think the authors should comment on the spatial resolution when using the phased-shaped beam and if/how this can be improved.

In relation to the chosen nanorod structure, it is unclear to me why the nanorod dimer (shown in Supplementary Info D) was not experimentally realized and studied. Such a system could really show the strength of using a phased-shaped beam (as also argued and illustrated by the simulations shown by the authors). Also, the nanorod dimer should not pose additional fabrication difficulties compared to the rod structures already prepared.

Overall, I find the manuscript quite unpolished. I think more effort can be spent on Figures 2, 3 and 4 (they even lack scale bars and color bars) to make them easily understandable and more appealing. I also suspect that the orientation of the rod in Fig 1(a) second column is wrong. In addition, I encountered quite a number of wordings which need to be edited, such as "visible light" (line 20), introduction of abbreviation PINEM (line 32) and not using it again, "synthetic expression" (line 58) and so on. Also, what does Eq. (4) mean?

Besides the above-mentioned issues, I list some more specific questions below

- On line 47 the authors state "it is impossible to distinguish modes that are degenerate in energy and space". What does this mean? Which modes are simultaneously degenerate in energy and space, yet not the same?
- On line 49-50 the authors hint to that using phase-shaped beams may even be more accurate than simulations in deducing the symmetry of a given mode. This seems like an exaggeration. Obviously, eigenmode calculations can straightforwardly be used to deduce the symmetry (as also done by the authors).
- Fig 3a shows the samples investigated by the authors. What is the purpose of the different orientations of the rods? Why not just use one orientation and then rotate the sample in the TEM? Also, what are the rod diameters/widths?
- For the results in Fig 3d, the authors state that a 50 nm probe size was used for the shaped beam. Is this also true in the normal-beam case?

Response to Reviewers' comments:

Reviewer #1:

This paper reports on the use of phase-shaped electron beam to selectively probe by TEM the symmetries of plasmonic resonances in metallic nanostructures. The main idea of the paper consists creating a 2-lobes electron beam whose phase symmetry allows discriminating the symmetries of the surface plasmon resonances (SPR) of a given metallic nano-object. To achieve the phase shaping of the incident electron beam, the authors use a ferromagnetic nanowire located in the condenser aperture. Such very simple device, similar to one already the used by the group in Antwerp for EMCD studies, allows creating two lobes beams with almost π phase shift difference between them. The interaction between this two phase-shifted lobes beam with the SPR is either constructive or destructive depending on the relative phases of the beam and the SPR symmetry. This simple and smart idea is analogous to polarised light / SPR interactions but offers the advantage to locally probe the SPR symmetry of individual object, thanks to the spatial resolution of TEM,

We thank reviewer #1 for appreciating the conceptually simple, yet innovative, nature of our work, and its relevance.

The paper clearly describes the method both theoretical and experimentally. Some points however are not so clear and could be improved for the benefit of the reader.

• The paper is first introduced by a schematic view of the experiment (fig. 1), which illustrates the effects of a conventional and the 2-lobed beam when interacting with the dipolar plasmonic resonance of a metallic nanorod. The illustration regarding the case where the 2-lobed beam is set perpendicular to the main axis of the Al nanorod is not so clear for me as I would expect almost no interaction with the dipolar SPR. Does the result presented in Fig. 1 (d) (right column) correspond to the interaction of the 2-lobed beam with the quadripolar SPR of the nanorod (which is not represented neither in the figure nor in the text)? If yes I suggest the authors to clarify this point, if not I also suggest the authors to explain what is the expected result of the 2-lobed beam with the dipolar SPR in such configuration.

The figure and caption are correct, as the scheme represents the interaction of a 2 lobed beam perpendicular to the nanorod. The beam does indeed interact with the particle and gets scattered, but due to interference effects the scattering only happens off-axis. This can be seen in the figure by considering that the various “lobes” seen in the third column of figure 1c interfere destructively, and the selectivity stems by only detecting electrons scattered at $k_{\perp}=0$. We see however that in the figure we are not entirely clear in this respect, and sought to make it clearer by fine tuning the relative size of a few elements (detectors “holes” are now smaller, patterns in fig.1d bigger) and by adding this sentence to the caption:

A spectrometer only accepting electrons scattered in the centre of the detection plane (dark ellipses at the bottom) is then sensitive only to plasmon modes which match the symmetry of the probe (second column), and ignores the other cases (first and third column) where electrons are only scattered off axis.

This is a manifestation of a more general feature of electron beam shaping. Here we use phase manipulation as an analogue of what polarisation is in light optics. However while a change in light polarisation will change the total cross section, just changing the electron's phase will change just the far field distribution of the scattered electrons but not the total

cross section. We have clarified the issue in the paper by adding the following discussion at the end of the theory section, right after the interferometric interpretation of the experiment: *It is important to point out that without a projective measurement, that is, a limited collection angle, the selectivity disappears. Transitions to tilted plane waves, as noted above, are still allowed, and in general for electron beams a change in the phase alone will not modify the total cross-section but only the angular distribution of scattered electron, in contrast to what happens in light optics upon a change in polarisation. A beam with identical intensity distribution to the two-lobed beam of eq. 3, but without the sign change ($\psi_{i\perp}=|x|\cdot\exp((x^2+y^2)/w^2)$) would result in an identical total cross section if all possible final states are allowed. It is only by performing an angular post-selection (i.e. measuring on axis) that the two beams give different selection rules.*

- *The 2-lobed beam is created thanks to the use of magnetic nanowires magnetized along the wire direction resulting in a phase shift between the beams passing on each side of the wire thanks to the Aharonov–Bohm effect. In addition to the phase shift (which depends on the magnetic induction within the wire), the authors evidence an intensity difference between these two beams and claim it comes from a non perfectly equal to π phase shift (0.87π). This is not straightforward for me, as we do not expect to observe any difference in the intensity of the two parts of the beam whatever the phase shift between them. Could the intensity difference originate from a slight shift of the incident beam with respect to the magnetic nanowire above the condenser aperture? Can the authors also clarify this point.*

This is because the electron probe directed on the sample is the Fourier transform of the wave function in the condenser aperture. Therefore, any change in the phase structure directly distorts the intensity distribution. To clarify the issue, we have included simulated wave functions for different values of phase shift in the supplementary information, and changed the main text accordingly. The point where the asymmetry is first mentioned now reads: *Since the plane where the aperture is placed is optically reciprocal to the sample plane, imperfections in the desired phase profile also cause the intensity distribution to change. Due to the phase difference not being exactly π radians, the beam is slightly asymmetric and presents a more intense and a weaker lobe, as well reflected by numerical simulation for the experimental value of the phase difference (fig. 2c) (see also the Supplementary Information).*

- *In figure 3, the authors present experimental and simulated EELS spectra obtained on dipolar and quadrupolar SPR of nanorods whose orientation is varied compared to the axis of the 2-lobed shaped beam (blue curves) and for normal round beam (red curves). The experimental spectra are very convincing on the influence of the beam symmetry with respect to the symmetry of the SPR. However, the simulated plot with a “normal” beam only reveals the quadrupole mode at 2.6 eV (dashed blue plot) while the dipolar one at 1.4 eV is also experimentally observed and expected?*

This is due to the fact that we simulated an infinitely small collection angle, and therefore have perfect selectivity, while in the real experiment the finite collection angle yields imperfect selection.

The fact that we do detect a bit of dipolar signal with the normal beam (solid blue plot) but no quadrupolar signal with the modified one (solid red plot) can be attributed to the fact that the dipolar mode normally gives a much stronger signal (see for instance *Rossouw et al. Nano Lett., 2011, 11 (4), pp 1499–1504*).

We took this into account in the text, by including this sentence in the paper:

As can be observed, the two spectra display strong selectivity in agreement with the above arguments, with the imperfect selection being due to the finiteness of the collection angle.

And this one in the caption to figure 3, also in reference to the comment of reviewer #2:

The spectra have been scaled for ease of visualisation as the blue spectra were significantly less intense, in agreement with the expected lower intensity of a quadrupolar peak.

- *At the end of the paper the authors show by simulation how a 2-lobed beam and even a 4-lobed beam could be efficient for probing the degenerated symmetries of SPRs in high symmetry particles like metallic square platelet (figure 5) or to analyse the coupling of plasmon modes in neighbouring nanoparticles. As a practical point of view and by curiosity, I wonder how the authors plan to set a 4-lobed beam as crossing 2 magnetic nanowires in the aperture condenser will result by dipolar magnetic interaction in a magnetic configuration quite far from the simple addition of two independent magnetic nanowire magnetized along their axis?*

In a nanowire the dominating magnetic energy term, is the demagnetisation energy / shape anisotropy, that the system minimises by orienting the magnetisation along the needle's long direction. That we are in this regime, is strongly suggested by the results in *Béché et al. Ultramicroscopy, in press* (available at <http://dx.doi.org/10.1016/j.ultramic.2016.05.006>) where the needles, despite being manufactured from a polycrystalline film, appear to be single domain and do not show stray field at the sides.

The configuration we seek from a crosswire structure would satisfy the requirement of minimising the surface energy, and similar (though not quite identical) structures have been demonstrated before (for instance *Pulecio, J. F. and Bhanja, S., Journal of Applied Physics, 107, 034308*).

If however this were to fail, it is still possible to mount two wires on the different sides (top and bottom) of the film constituting the aperture, putting a distance between the two that can go up to several microns, thus significantly limiting the interaction.

To summarize, the main purpose of the paper demonstrating a smart way to probe the symmetries of SPR as well as its experimental and theoretical support largely deserve to be published in Nature Communications after the few previous points have been discussed and corrected.

We thank reviewer #1 for her/his kind remarks, and hope we have answered satisfactorily to all his/her observations.

Reviewer #2 (Remarks to the Author):

In their manuscript entitled "Probing the symmetry and phase of localised surface plasmon resonances with phase-shaped electron beams", Guzzinati et al demonstrate theoretically and experimentally that using phase-shaped electron beams in a TEM allows detecting specific surface plasmon modes based on their symmetry. Although the concept bears great similarity with the use of polarized light in optical microscopes to selectively excite specific plasmon modes, the presented concept goes beyond as it adds the nanoscale spatial resolution of TEM.

This work brings important and new results. The paper is extremely well written. In particular, the introductory discussion and presentation of this new concept is very clear. Both theoretical and experimental results are very convincing. The results

presented in Figure 4 clearly lie at the limit of what can be achieved today on state of the art Transmission Electron Microscopes.

We thank reviewer #2 for appreciating both the novelty and the importance of our work, as well as the experimental challenges we had to overcome in the present work.

In conclusion, this paper proposes and demonstrates a new concept that will undoubtedly be of the highest interest for the nano-optics community. The presentation of the results and their discussion is very clear. I strongly recommend this paper for publication in Nature Communications.

We thank reviewer #2 for his/her kind remarks, and trust she/he will find the following answers satisfactory.

I have two comments:

1) Contrary to what is suggested in the title, the phase of the plasmon modes is only indirectly probed in the described experiments. In their work, the authors demonstrate that an a-priori knowledge of the symmetries of the plasmon modes supported by a metal nanoparticle allows choosing the electron beam shape that can selectively detect a given mode. However, the presented technique does not yield a quantitative measurement of the phase of the plasmon mode but rather gives access to its symmetry which is already an excellent step forward. Using the analogy with atomic physics, information about the symmetry of the coupling operator can be gained from the selection rules but no information about the phase.

We thank the reviewer for this very relevant comment. We have now dropped the word phase, and the new title we propose is:
“*Probing the symmetry of the potential of localised surface plasmon resonances with phase-shaped electron beams*”.

2) In the present work, the authors show how the introduction of a phase jump in the wavefunction of an electron beam can excite selected plasmon modes. This strategy can be compared to the introduction of phase jumps in the incident electric field driving a gap nanoantenna as proposed in Volpe et al, Nano Lett., 2009, 9 (10), pp 3608–3611. In this work, high order Hermite-Gaussian beams have been used to switch on and off the hot spots of a linear dimer gold nanotenna. This work could be added as a reference.

We thank reviewer #2 for bringing this to our attention. We have added a citation to this in the introduction:

While beam shaping has successfully been used in light optics to offer an extra degree of control over the response of plasmonic systems (see [47] for an interesting parallel to the present work), the spatial resolution remains limited.

3) The quality of some figures could be improved (fonts homogenized for instance)

We have now modified all figures to use the same typeface (and in similar sizes). In that respect, please notice that we envision figure 1 as being a two-column figure, and all others as one-column, hence the difference in font sizes.

4) Line 281: *Are the spectra normalized before or after substration of reference zero loss peaks ?*

The spectra had been normalised before, but in figure 3 they have actually been scaled to have the same height for ease of visualisation, as the blue spectra would have significantly less strong (compared to the total dose contained in the zero loss peak), so we effectively undid the previous normalisation. That higher order modes give weaker signal is entirely expected, as also highlighted by our comment to referee #1.

I have modified the caption of figure 3 to mention the scaling:

The spectra have been scaled for ease of visualisation as the blue spectra were significantly less intense, in agreement with the expected lower intensity of a quadrupolar peak.

Additional remarks:

- 1) *Figure 1 - caption: the height in the 3D representation is proportional to the modulus squared of psi, isn't it ?*
- 2) *Figure 1 - caption: it would be better to use the same font for F as in the figure.*
- 3) *Line 43 : "electric field"*
- 4) *Line 80: Is it "present" or "presented" ?*
- 5) *Figure 5 - Caption: "with our modified and corresponding". A word is missing*
- 6) *Line 210: "for two different orientationS". An "s" is missing.*
- 7) *Line 234: "Are appear very".*
- 8) *Line 285 and 287 have a repetition "using the freely available simulation code MNPBEM"*
- 9) *Line 288: "eigenmodes and spectral functions". An "s" is missing.*
- 10) *Line 428: "we need a to specify"*
- 11) *Line 431: "the opticAL axis"*
- 12) *Line 433: "we can sum, and due to the non-recoil approximation, resul...". This sentence should be improved.*

We thank the reviewer for pointing these out these mistakes. We have now corrected all of them.

Reviewer #3 (Remarks to the Author):

The authors study the plasmonic response of aluminum nanorods using electron energy-loss spectroscopy (EELS) in a modified transmission electron microscope. By generating an electron beam with a Hermite-Gaussian phase distribution, the authors are able to selectively excite plasmonic modes of certain symmetry. In addition, the authors can also retrieve phase information of the plasmonic mode, which is not possible with standard EELS (i.e., non-phased shaped beams).

I find the idea of using phase-shaped electron beams in EELS for optical studies very promising and appealing for a broad range of applications. I applaud the authors for the significant novel improvement on this topic shown in their manuscript, but I find the experimental results slightly demotivating.

We thank reviewer #3 for pointing out the novelty of our work, and the promise it holds. While we do recognise that the experimental data are noisy, what we are showing is at the very limit of what is currently possible, even with state of the art instrumentation, as noted by

reviewer #2. Two examples: it's currently impossible to achieve sufficiently small collection angles without compromising the spatial resolution, and it's impossible to achieve the high coherence needed for beam shaping, without reducing the beam current far below what is commonly used in spectroscopic experiments.

Also we think that the experimental data we show, when considered in its entirety, forms a coherent picture that strongly supports the model we propose.

I think this is mainly rooted in the nanorod structure, which was chosen by the authors to highlight the benefit of phased-shaped beams. While Fig. 3d shows that a shaped beam and a normal beam give rise to resonances at different energies (due to excitation of dipole and quadrupolar modes, respectively) for one particular position, the data shown in Figure 4 is less convincing (especially Fig. 4c). Given the quite large size of the nanorod in Fig. 4, the data suggests that when employing a phase-shaped beam you lose the beneficial spatial resolution of standard EELS (which is also praised by the authors in their introduction). I think the authors should comment on the spatial resolution when using the phased-shaped beam and if/how this can be improved.

The limited resolution is not due to the use of phase shaped beams, but to the limits of the current generation of instrumentation. In other words it's a technological limit, not a physical one.

The limited collection angle we require in order to have the selective detection cannot be achieved in the conventional operation mode of the microscope. This forces us to switch off the objective lens to achieve sufficiently small angles, and unsurprisingly this also compromises the spatial resolution we can achieve. At present the best option is to perform a 2-step experiment: (i) use conventional STEM-EELS to map the spatial distribution of the plasmon with the usual high-spatial resolution, and (ii) use beam shaping to "just" fill in the missing information, i.e.: the symmetry and signs.

It could also be argued that in this particular experiment, the highest resolutions are also not desirable since symmetry, or even local symmetry, is a property that characterises an extended region of the plasmon, and therefore is less relevant on a smaller scale.

The current spatial resolution, though limited, is still beyond the capabilities of optical techniques.

In relation to the chosen nanorod structure, it is unclear to me why the nanorod dimer (shown in Supplementary Info D) was not experimentally realized and studied. Such a system could really show the strength of using a phased-shaped beam (as also argued and illustrated by the simulations shown by the authors). Also, the nanorod dimer should not pose additional fabrication difficulties compared to the rod structures already prepared.

We performed and included the simulations, as a speculation on the possible applications and future directions of investigation. With this paper however, we want to focus on the proof of principle, that is on introducing and demonstrating the new physics described here. We feel that to this purpose the current data are convincing, and we are eager to share our results with the scientific community.

Overall, I find the manuscript quite unpolished. I think more effort can be spent on Figures 2, 3 and 4 (they even lack scale bars and color bars) to make them easily understandable and more appealing.

As can be seen by our replies, we have addressed and improved many parts of the manuscript, which we feel is now in better shape. We have polished and added colour bars and scale bars to figure 2, 3 and 4.

I also suspect that the orientation of the rod in Fig 1(a) second column is wrong. In addition, I encountered quite a number of wordings which need to be edited, such as “visible light” (line 20), introduction of abbreviation PINEM (line 32) and not using it again, “synthetic expression” (line 58) and so on. Also, what does Eq. (4) mean?

We thank the referee for pointing these out. We inverted the rod’s orientation in figure 1(a), we more clearly specified the meaning of visible light by specifying the wavelength range, removed the abbreviation PINEM, and replaced “synthetic” with “concise”. In equation (4) *cst* was used as an abbreviation for *constant* without specifying it. We have corrected this by using the whole word.

Besides the above-mentioned issues, I list some more specific questions below

- *On line 47 the authors state “it is impossible to distinguish modes that are degenerate in energy and space”. What does this mean? Which modes are simultaneously degenerate in energy and space, yet not the same?*

One obvious example would be the two dipolar modes of the square shown in figure 5. They have the exact same energy, and the charge peaks in the same points. Our wording was however inaccurate, and we have now corrected it to:

“Furthermore, it is impossible to distinguish modes whose electrical field modulus ($|E_z|$) and energy are near degenerate”

- *On line 49-50 the authors hint to that using phase-shaped beams may even be more accurate than simulations in deducing the symmetry of a given mode. This seems like an exaggeration. Obviously, eigenmode calculations can straightforwardly be used to deduce the symmetry (as also done by the authors).*

We understand that the wording was confusing, we have now modified the sentence:

In the lower symmetry cases, even the simulations become of limited help as the computation time increases greatly.

- *Fig 3a shows the samples investigated by the authors. What is the purpose of the different orientations of the rods? Why not just use one orientation and then rotate the sample in the TEM? Also, what are the rod diameters/widths?*

Having different orientations available markedly simplified the experiment, saving us the inconvenience of needing to rotate the actual sample.

All nanorods are about 50 nm wide, as we have now noted in the “Methods” section.

- *For the results in Fig 3d, the authors state that a 50 nm probe size was used for the shaped beam. Is this also true in the normal-beam case?*

Experimentally switching between the two beams only corresponds to switching between two apertures of identical size, so the angles involved are identical in the two cases. The shaped

beam is elongated, having a size of $\sim 50\text{nm}$ in the long direction, and a size of $\sim 30\text{nm}$ in the short one. The normal beam generated with the same convergence angle has a size of $\sim 30\text{nm}$.

REVIEWERS' COMMENTS:

Reviewer #1 (Remarks to the Author):

Second Report on: "Probing the symmetry and phase of localised surface plasmon resonances with phase-shaped electron beams » by Giulio Guzzinati, Armand Béch , Hugo Lourenço- Martins, Jérôme Martin, Mathieu Kociak, and Jo Verbeeck

Ref: # NCOMMS-16-24651-T

The points that were unclear for me have been clarified by the authors. The manuscript has therefore been improved.

The paper then clearly describes the original method proposed both theoretical and experimentally.

It then deserves to be published in Nature Comm.

Reviewer #2 (Remarks to the Author):

The authors have fully answered the points raised in my first report. This study clearly presents a new concept which goes beyond the state of the art in electron spectroscopy of plasmonic nanostructures. This new approach will undoubtedly stimulate a wealth of investigations. To my opinion, this work deserves to be published in Nature Communications. I strongly recommend this manuscript for publication.

Additional remarks:

p12 - L232 : "symmetry arguments Figure 1" - missing word
p15-L290 " on an FEI"

Reviewer #3 (Remarks to the Author):

I have carefully read the authors replies to my comments and the comments of the other reviewers. I find that the authors have satisfactorily answered all of my concerns. I therefore recommend the paper for publication in Nature Communications. The work is very novel and will be of great interest to the community.

A minor thing: in the reply to one of my comments, the authors stated that they changed the orientation of the nanowire in the second column of Figure 1a. However, I don't see this change in the revised manuscript. The schematic of the wire in the second of Figure 1a is still rotated wrongly. I believe it should be parallel to the plasmon electric potential shown in Fig 1b of the same column.

REVIEWERS' COMMENTS:

Reviewer #1 (Remarks to the Author):

The points that were unclear for me have been clarified by the authors. The manuscript has therefore been improved.

The paper then clearly describes the original method proposed both theoretical and experimentally.

It then deserves to be published in Nature Comm.

We are glad we have been able to clear the issues.

Reviewer #2 (Remarks to the Author):

The authors have fully answered the points raised in my first report. This study clearly presents a new concept which goes beyond the state of the art in electron spectroscopy of plasmonic nanostructures. This new approach will undoubtedly stimulate a wealth of investigations. To my opinion, this work deserves to be published in Nature Communications. I strongly recommend this manuscript for publication.

Additional remarks:

*p12 - L232 : "symmetry arguments Figure 1" - missing word
p15-L290 " on an FEI"*

We have corrected these points. Thanks for pointing them out.

Reviewer #3 (Remarks to the Author):

I have carefully read the authors replies to my comments and the comments of the other reviewers. I find that the authors have satisfactorily answered all of my concerns. I therefore recommend the paper for publication in Nature Communications. The work is very novel and will be of great interest to the community.

We thank Reviewer #3 for appreciating the novelty of our work.

A minor thing: in the reply to one of my comments, the authors stated that they changed the orientation of the nanowire in the second column of Figure 1a. However, I don't see this change in the revised manuscript. The schematic of the wire in the second of Figure 1a is still rotated wrongly. I believe it should be parallel to the plasmon electric potential shown in Fig 1b of the same column.

This is subtle. We are sure that the direction of the wire is correct. It is, and should be, parallel to the zero-line in the wave-function of the second and third column of figure 1a, and not to the potential of figure 1b. The wire should indeed be 90 degrees rotated between the two cases.

In our reply, we referred to the fact that we have “reversed” the wire compared to our initial submission: we have swapped the red and blue end, as there was an inconsistency between column 2 and 3, which is now solved.